# Synthetic Amphipathic β-Sheet Temporin-Derived Peptide with Dual Antibacterial and Anti-Inflammatory Activities

**DOI:** 10.3390/antibiotics11101285

**Published:** 2022-09-21

**Authors:** Rosa Bellavita, Elisabetta Buommino, Bruno Casciaro, Francesco Merlino, Floriana Cappiello, Noemi Marigliano, Anella Saviano, Francesco Maione, Rosaria Santangelo, Maria Luisa Mangoni, Stefania Galdiero, Paolo Grieco, Annarita Falanga

**Affiliations:** 1Department of Pharmacy, University of Naples “Federico II”, Via Montesano 49, 80131 Naples, Italy; 2Laboratory Affiliated to Istituto Pasteur Italia-Fondazione Cenci Bolognetti, Department of Biochemical Sciences, Sapienza University of Rome, 00185 Rome, Italy; 3Alsa Lab SAS., Consorzio Fracta Labor Area PIP Lotto 25, Frattamaggiore, 80027 Naples, Italy; 4Department of Agricultural Science, University of Naples “Federico II”, 80055 Portici, Italy

**Keywords:** Temporin L, physical-chemical property, peptide hydrophobicity, antimicrobial peptides, anti-inflammatory activity

## Abstract

Temporin family is one of the largest among antimicrobial peptides (AMPs), which act mainly by penetrating and disrupting the bacterial membranes. To further understand the relationship between the physical-chemical properties and their antimicrobial activity and selectivity, an analogue of Temporin L, [Nle^1^, dLeu^9^, dLys^10^]TL (Nle-Phe-Val-Pro-Trp-Phe-Lys-Phe-dLeu-dLys-Arg-Ile-Leu-CONH_2_) has been developed in the present work. The design strategy consisted of the addition of a norleucine residue at the N-terminus of the lead peptide sequence, [dLeu^9^, dLys^10^]TL, previously developed by our group. This modification promoted an increase of peptide hydrophobicity and, interestingly, more efficient activity against both Gram-positive and Gram-negative strains, without affecting human keratinocytes and red blood cells survival compared to the lead peptide. Thus, this novel compound was subjected to biophysical studies, which showed that the peptide [Nle^1^, dLeu^9^, dLys^10^]TL is unstructured in water, while it adopts β-type conformation in liposomes mimicking bacterial membranes, in contrast to its lead peptide forming α-helical aggregates. After its aggregation in the bacterial membrane, [Nle^1^, dLeu^9^, dLys^10^]TL induced membrane destabilization and deformation. In addition, the increase of peptide hydrophobicity did not cause a loss of anti-inflammatory activity of the peptide [Nle^1^, dLeu^9^, dLys^10^]TL in comparison with its lead peptide. In this study, our results demonstrated that positive net charge, optimum hydrophobic−hydrophilic balance, and chain length remain the most important parameters to be addressed while designing small cationic AMPs.

## 1. Introduction

The growing public health threat of antimicrobial resistance (AMR) imposes a great challenge to the scientific community for the development of alternative strategies to the traditional small molecule therapeutics [1,2]. This global effort has resulted in significant interest in antimicrobial peptides (AMPs) and significant advances in our understanding of their antimicrobial activity at the molecular scale [3,4,5].

AMPs are less than 50 amino acids in length and have been found in microorganisms and humans [6,7]. They are generally cationic and amphipathic, although their sequences and secondary structures widely differ among species [8]. AMPs act as the first line of the innate defense system, showing multiple antimicrobial features, including potent, rapid, and broad-spectrum antimicrobial and anti-biofilm activities as well as delayed promotion of antimicrobial resistance [9,10,11,12]. This is due to their multiple modes of action, such as membrane disruption, membrane permeabilization followed by intracellular mechanisms and pathways, and immunomodulatory activities, in contrast to the specific targets of the antibiotics [13,14].

AMPs are generally composed of cationic, hydrophilic, and hydrophobic amino acids arranged in an amphipathic structure [15]. Cationic amino acids (lysine/arginine) provide the initial electrostatic interaction with the negatively charged membrane surfaces of the microbes [16]. Aromatic residues such as phenylalanine and tryptophan play important roles in antimicrobial and hemolytic activities as they facilitate the formation of amphipathic structures and are fundamental for the interaction at the interface between the aqueous solution and the hydrophobic membrane bilayer [17,18].

Thereby, the action of AMPs results to be a combination of electrostatic and hydrophobic interactions with a membrane, consequently. For more neutral membranes, where electrostatic peptide–lipid interactions are minimized, the hydrophobic effect plays a major role related to hosting cell toxicity [19]. In fact, the increase of the hydrophobic interaction is strongly correlated with mammalian cell toxicity and loss of antimicrobial specificity [20]. Mainly, selective antimicrobial activity results from a delicate balance of hydrophobic and electrostatic interactions between the AMP and the target membrane [21].

Another key feature that is recently gaining much attention is the intrinsic self-assembly of AMPs [22,23]. AMPs affect the microbial membrane, either by lipid phase destabilization or pore formation, eventually leading to cell lysis and death, and their amphipathic structure is responsible for intrinsic self-assembly, while hydrophobic and cationic charges ensure interactions with phospholipids and membrane insertion leading to antimicrobial activity [24,25]. Hence, the amphipathic structure is key to drive self-assembly in membrane environments while ensuring activity. By introducing rational modifications in the amphipathic structure such as the incorporation of long lipid tails, it is possible to fine-tune self-assembly into supramolecular structures of different morphologies while conserving antimicrobial activity [26,27].

With increasing insight into the parameters responsible for activity and selectivity, we are attempting to engineer AMPs, with the aim to improve the physio-chemical properties and increase the immunomodulatory efficacy.

Useful approaches to enhance peptide stability and effectiveness in vivo consists of peptide modifications and/or the development and application of drug delivery system [28,29,30,31]. Among conventional peptide modifications, including amidation (*C*-terminal), acetylation (*N*-terminal), methylation, and cyclization strategies, are frequently used to increase resistance to peptidases and proteases [32]. AMPs with *C*-terminal amidation are common in nature, including those belonging to the temporin family, widely studied by our group [33,34]. We have particularly focused on Temporin L (TL, FVQWFSKFLGRIL), one of the most potent temporin isoforms but also the most highly hemolytic [35]. Indeed, our previous attempts were aimed at improving its biological profile through the application of several synthetic strategies, including proline decoration [36], lipidation, and side chain-to-side chain cyclizations [37,38,39,40]. Several TL analogues were developed and characterized through biophysical techniques to elucidate their mode of action, some of them also displayed a wide broad-spectrum antibacterial activity and no cytotoxicity on human cells. In our previous studies, by the incorporation of a d-Lys residue in position 10 of the helical region of TL analogue [Pro^3^, dLeu^9^]TL [41], we discovered another analogue, namely [dLeu^9^, dLys^10^]TL (peptide sequence: FVPWFSKF*lk*RIL), which exhibited a remarkable activity against several bacterial strains, even including yeasts and Gram-negative strains [41,42]. In addition, due to the absence of cytotoxicity, this peptide was also tested in vivo, showing an anti-inflammatory activity in response to zymosan-induced peritonitis [43].

Here, we evaluated the role of the peptide hydrophobicity introducing a residue of norleucine (Nle) in the flexible *N-*terminal region of [dLeu^9^, dLys^10^]TL, exploring how variations of the hydrophobic−hydrophilic balance can influence the peptide biological profile. The broad-spectrum antimicrobial activity of the designed analogue [Nle^1^, dLeu^9^, dLys^10^]TL was evaluated against Gram-positive and Gram-negative strains and the cytotoxicity was assessed on erythrocytes and human keratinocytes. The mechanism of action was studied in detail and compared to the lead temporin analogue by using various biophysical and spectroscopic analysis. In addition, the anti-inflammatory activity of [Nle^1^, dLeu^9^, dLys^10^]TL was explored on murine macrophage cells line by evaluating the effect on interleukin-6 (IL-6) production.

## 2. Results

### 2.1. Peptide Design

One of the main features of AMPs is their hydrophobic−hydrophilic balance which is directly correlated to their antimicrobial activity. We have previously developed the temporin analogue [dLeu^9^, dLys^10^]TL with decreased hemolytic activity and enhanced antimicrobial activity against both Gram-positive and Gram-negative bacteria [41,42]. Here, we decided to analyze the crucial role played by hydrophobicity by adding a norleucine (Nle) residue in the flexible *N*-terminal region of [dLeu^9^, dLys^10^]TL, and its intrinsic self- capabilities due to the short chain of carbons present on Nle. Nle is an isomer of natural amino acids leucine and isoleucine featured by an unbranched side chain unlike these latter, which may facilitate the incorporation into the phospholipid bilayer of the bacterial membrane [44,45].

### 2.2. Physicochemical Properties

We probed the influence of Nle on the physicochemical properties of peptide [Nle^1^, dLeu^9^, dLys^10^]TL, including hydrophobic moment and hydrophobicity, which are crucial elements for the antimicrobial activity [15].

The hydrophobic moment (μH) measures the amphiphilicity of a peptide sequence [46]. AMPs typically present a high hydrophobic moment and moderate hydrophobicity [47]. Both peptides, [dLeu^9^, dLys^10^]TL and [Nle^1^, dLeu^9^, dLys^10^]TL, have high μH values of 0.670 and 0.601 (obtained from the HeliQuest website), respectively, demonstrating the maintenance of the amphiphilic nature and the remarkable correlation with their strong antimicrobial profile.

The peptide hydrophobicity was measured by using the GRAVY (Grand Average of Hydropathy) index that calculates the hydrophobicity of a peptide obtained by the sum of the hydropathy values of all the amino acids divided by the sequence length [48]. The positive values of the GRAVY index obtained for peptides [dLeu^9^, dLys^10^]TL and [Nle^1^, dLeu^9^, dLys^10^]TL from the ExPASy website further supports their hydrophobic nature (Table 1). Interestingly, when Nle residue was added to the N-terminus of [dLeu^9^, dLys^10^]TL, the GRAVY index increased from 0.700 ([dLeu^9^, dLys^10^]TL) to 0.921 for [Nle^1^, dLeu^9^, dLys^10^]TL, indicating an enhancement of peptide hydrophobicity. Likewise, the aliphatic index, defined as the relative volume occupied by aliphatic side chains, increased for peptide [Nle^1^, dLeu^9^, dLys^10^]TL obtaining a higher value of 132.14 than the value of 112.31 calculated for its lead peptide.

Another tool to estimate peptide hydrophobicity and how the hydrophobic amino acids contribute to the repartition of peptides at the membrane interface is represented by the Wimley–White hydrophobicity scale [49,50]. This parameter indicates the free energy associated with the repartition of a peptide from an aqueous environment to the hydrophobic environment such as octanol. We calculated positive Gibbs energy values of +4.15 Kcal∙mol^−1^ and +2.90 Kcal∙mol^−1^ for peptides [dLeu^9^, dLys^10^]TL and [Nle^1^, dLeu^9^, dLys^10^]TL, respectively (Table 1). These data obtained from the *PepDraw* website indicate a localization of peptides in the interfacial region of membranes.

### 2.3. Broad-Spectrum Antimicrobial Activity of [Nle^1^, dLeu^9^, dLys^10^]TL

Minimum inhibitory concentrations (MICs) of [Nle^1^, dLeu^9^, dLys^10^]TL were determined against Gram-positive and Gram-negative strains. Previously, we demonstrated that the antimicrobial activity of [dLeu^9^, dLys^10^]TL is targeted at the cell membrane of both Gram-positive and Gram-negative bacteria and is able to eradicate selected bacteria at a relatively low concentration varying in the range 6–12 μM for Gram-negative bacteria and 3–6 μM for Gram-positive bacteria (Table 2) [27]. The analogue [Nle^1^, dLeu^9^, dLys^10^]TL bearing Nle in *N-*terminus showed an enhanced activity against *A. baumannii* ATCC 19606 (MIC = 3.12 μM) and *E. coli* ATCC 25922 (MIC = 6.25 μM) in comparison with [dLeu^9^, dLys^10^]TL (MIC = 6.25 and 12.5 μM, respectively).

In addition, [Nle^1^, dLeu^9^, dLys^10^]TL maintained a good activity against *P. aeruginosa* ATCC 27853 and *K. pneumoniae* ATCC BAA-1705 with the same MIC = 12.5 μM of [DLeu^9^, DLys^10^]TL. Furthermore, the incorporation of the residue Nle promoted an increased activity against the Gram-positive bacteria *S. aureus* ATCC 25923 and *B. megaterium Bm11* with MIC of 3.12 and 0.78 μM, respectively, while the MIC against *S. epidermidis* ATCC 12228 was equal to 3.12 μM, as for the lead peptide.

### 2.4. [Nle^1^, dLeu^9^, dLys^10^]TL Was Stable and Did Not Cause Cytotoxicity on Human Cells

The hemolytic and cytotoxic activities of peptide [Nle^1^, dLeu^9^, dLys^10^]TL were investigated on red blood cells and human keratinocytes, respectively. Before the evaluation of toxicity profile, firstly, we determined the proteolytic degradation of peptide [Nle^1^, dLeu^9^, dLys^10^]TL (Figure 1, panel A). The peptide [Nle^1^, dLeu^9^, dLys^10^]TL was stable up to 45 min (~75% of intact peptide), while a partial degradation was observed starting from 90 min, detecting ~50% of the intact peptide within 120 min.

At this stage, we investigated the hemolytic effect of peptide [Nle^1^, dLeu^9^, dLys^10^]TL on red blood cells after a treatment of 40 min (Figure 1, panel B). The peptide showed a weak hemolytic effect at the concentrations of 3.12 μM and 6.25 μM (lower than 20%), while the strongest hemolytic effect was observed at the highest concentrations of 50 μM and 100 μM (~40% of hemolysis). The hemolytic profile of peptide [Nle^1^, dLeu^9^, dLys^10^]TL resulted similar to that of its lead peptide [dLeu^9^, dLys^10^]TL already reported in Bellavita et al. [27].

Similarly, the peptide [Nle^1^, dLeu^9^, dLys^10^]TL caused only a slight reduction in human keratinocytes viability in the range 3.12–12.5 μM after incubation for 2 h and 24 h (Figure 1, panel C), while it became cytotoxic at 25 μM (~40% and ~50% of cell viability at 2 h and 24 h, respectively) and induced almost complete cell death at the highest concentrations (50 μM and 100 μM).

### 2.5. Aggregation Propensity of [Nle^1^, dLeu^9^, dLys^10^]TL in Large Unilamellar Vesicles (LUVs)

The impact of Nle on the aggregation propensity was studied in solution and in presence of large unilamellar vesicles (LUVs) mimicking Gram-positive and Gram-negative membranes. The aggregation in aqueous solution was monitored using Nile Red (NR) as a fluorescent probe [51]. Our previous studies on the aggregation mode of [dLeu^9^, vLys^10^]TL showed that the peptide is unable to aggregate in solution in the range of concentrations 1 μM–200 μM [27]. The introduction of the Nle residue conferred to peptide [Nle^1^, dLeu^9^, dLys^10^]TL a low propensity to aggregate in aqueous solution as shown by its high value of critical aggregation concentration (CAC) equal to 38.9 μM. This result suggests that the peptide is in the monomeric state in aqueous solution.

On the other hand, the increase of hydrophobicity due to the Nle residue promoted a faster tendency of the peptide [Nle^1^, dLeu^9^, dLys^10^]TL to aggregate in LUVs mimicking Gram-positive bacterial membrane (Figure 2). In this case, the probe Thioflavin T was used to monitor the peptide oligomerization in LUVs.

LUVs composed of DOPG/CL (58/42 ratio in moles) and DOPG/DOPE/CL (63/23/12 ratio in moles) were used to mimic Gram-positive and Gram-negative cell membranes, respectively. LUVs were titrated with different peptide concentrations of 5, 10, 15, 10, 30, and 50 μM. The peptide [Nle^1^, dLeu^9^, dLys^10^]TL was already oligomerized at a low concentration of 5 μM (~22% of aggregation) and it was completely aggregated at concentration of 30 μM. In contrast, the peptide [dLeu^9^, dLys^10^]TL progressively oligomerized in LUVs (DOPG/CL) as described in Bellavita et al. [27]; in fact, it was partially aggregated at a higher concentration of 20 μM (~56% of aggregation) and completely aggregated at the highest concentration of 50 μM [27].

Instead, both peptides showed a tendency to oligomerize progressively in the presence of LUVs (DOPG/DOPE/CL) mimicking Gram-negative membrane, and both were completely aggregated at the concentration of 20 μM (Figure 2, panel B).

### 2.6. [Nle^1^, dLeu^9^, dLys^10^]TL Forms β-Aggregates in LUVs

The molecular conformation of [Nle^1^, dLeu^9^, dLys^10^]TL was investigated by circular dichroism (CD) spectroscopy both in LUVs made of DOPG/CL and DOPG/DOPE/CL (Figure 3, panel A). Previously, we observed a high tendency of the peptide [dLeu^9^, dLys^10^]TL to form helical aggregates both in LUVs mimicking Gram-positive and Gram-negative membranes [27]. In contrast, the introduction of Nle induced changes in molecular conformation, determining the formation of β-aggregates. In fact, the CD spectrum showed a negative band at ~220 nm (Figure 3), indicating the β-type conformation for the peptide [Nle^1^, dLeu^9^, dLys^10^]TL, featured by alternating hydrophilic and hydrophobic amino acids (Figure 3, panel B).

### 2.7. [Nle^1^, dLeu^9^, dLys^10^]TL Changes Significantly the Membrane Fluidity

The changing of the membrane fluidity in presence of [Nle^1^, dLeu^9^, dLys^10^]TL was studied through a biophysical assay using LUVs mimicking Gram-positive and Gram-negative membranes. We used Laurdan as the fluorescent probe, whose emission can shift from 440 nm, indicating the LUVs ordered phase, to 490 nm indicating the LUVs disordered phase [52]. We calculated the Generalized Polarisation (GP) parameter which quantifies the change in the lipid fluidity in presence of peptide at two different concentrations of 5 μM and 30 μM (Table 3). In contrast to its lead peptide, [Nle^1^, dLeu^9^, dLys^10^]TL caused a significant increase of GP value already at the concentration of 5 μM, indicating the shift from a disordered state (−0.18) to an ordered phase (0.03). Moreover, as for its lead peptide [20], the highest increase of GP value (0.29) for peptide [Nle^1^, dLeu^9^, dLys^10^]TL was calculated at the concentration of 30 μM. Instead, both peptides showed the same behavior on membrane fluidity in presence of LUVs (DOPG/DOPE/CL) mimicking Gram-negative membrane. In this condition, the GP value significantly increased both at 5 μM and 30 μM, indicating the shift from a disordered state to an ordered phase for liposomes.

### 2.8. [Nle^1^, dLeu^9^, dLys^10^]TL Induces Leakage LUVs

Our previous studies showed the capacity of the lead peptide [dLeu^9^, dLys^10^]TL to induce a membrane perturbation followed by the bacterial death [27]. In this study, we evaluated the membranolytic activity of analogue [Nle^1^, dLeu^9^, dLys^10^]TL in presence of LUVs mimicking Gram-positive and Gram-negative membranes by using ANTS/DPX leakage assay. As shown in Figure 4, [Nle^1^, dLeu^9^, dLys^10^]TL induced higher leakage of LUVs mimicking Gram-positive (DOPG/CL) than its lead peptide. Indeed, after the treatment with 10 μM of [Nle^1^, dLeu^9^, dLys^10^]TL, we observed ~60% of LUVs leakage in comparison with ~23% of leakage induced by [dLeu^9^, dLys^10^]TL [27]. This is in line with the biological data of peptide [Nle^1^, dLeu^9^, dLys^10^]TL showing a greater activity against Gram-positive strains. Instead, similarly to its lead peptide, [Nle^1^, dLeu^9^, dLys^10^]TL promoted the partial leakage of LUVs mimicking Gram-negative (DOPG/DOPE/CL) even at highest concentration of 50 μM (~20% of leakage). These data suggest that the mode of action of both peptides may also involve a carpet-based effect or other molecular mechanisms underlying their activity against Gram-negative strains.

### 2.9. [Nle^1^, dLeu^9^, dLys^10^]TL Exhibited Anti-Inflammatory Activity In Vitro Model

Previously, we demonstrated that the peptide [dLeu^9^, dLys^10^]TL exhibited a significant anti-inflammatory activity in zymosan-induced peritonitis as an in vivo model and reduced the level of pro-inflammatory cytokines such IL-6, monocyte chemoattractant protein-1 (MCP-1), and tumor necrosis factor alpha (TNF-α) [43]. Here, we evaluated the impact of the introduction of the residue Nle on anti-inflammatory potency. Firstly, we evaluated the cytotoxic effect of [Nle^1^, dLeu^9^, dLys^10^] on murine macrophage cell line using [3-(4,5-dimethylthiazol-2-yl)-2,5-diphenyltetrazolium bromide] (MTT) assay (conducted in compliance with ISO 10993–5 (2009) in a concentration range 1 to 30 μM. As shown in Figure 5, our in vitro examination revealed a safe profile for the peptide at the concentrations of 1–3 μM, whereas only the highest concentrations of 10 and 30 μM caused a significant reduction of cell viability (dotted lines indicate 75% of cell viability).

These results have prompted us to investigate the potential anti-inflammatory activity of [Nle^1^, dLeu^9^, dLys^10^] at the concentration of 3 μM. As reported in Figure 6, stimulation of J774A.1 cells with lipopolysaccharide (LPS) (10 µg∙mL^−1^) induced a significant increase in IL-6 levels compared to the control (Ctrl) group. Interestingly, the peptide was able to modulate IL-6 production in macrophages stimulated with the pro-phlogistic agent (Figure 6).

## 3. Discussion

Several physicochemical features control the activity and the mechanism of action of AMPs [53,54]. A key feature is represented by the net positive charge of the AMP involved in electrostatic interactions with negatively charged phospholipids, but hydrophobicity also plays a crucial role because it determines the extent of membrane permeability of AMPs and drives their insertion and aggregation into lipid bilayers of the bacterial membrane [15,55]. In addition, another valuable feature of AMPs is represented by amphipathicity which favors the membrane binding by interacting with the hydrophobic−hydrophilic portions of the lipids. Several works showed that an increase of amphipathicity is correlated to an enhancement of antimicrobial activity [55].

Therefore, the application of any chemical modifications on AMPs sequence requires the maintenance of the hydrophobic–hydrophilic balance to obtain highly bacterial selective peptide therapeutics [56].

Herein, we explored the role of hydrophobicity on the activity and mode of action of the TL analogue, namely [dLeu^9^, dLys^10^]TL, through the incorporation of a norleucine residue in its flexible *N-*terminal region. Previously, this peptide resulted to be a promising drug candidate for its remarkable antimicrobial activity against some Gram-positive and Gram-negative strains and its very low cytotoxicity even at high concentrations. In this work, we increased the hydrophobicity of [dLeu^9^, dLys^10^]TL preserving the right hydrophobic–hydrophilic balance without increasing its cytotoxic profile. The increase of hydrophobicity of the peptide [Nle^1^, dLeu^9^, dLys^10^]TL was detected by a high GRAVY index value of 0.921 and its high aliphatic index of 132.14. These data demonstrated the contribution of Nle to the hydrophobicity of the lead peptide [dLeu^9^, dLys^10^]TL, which has lower values of GRAVY index and aliphatic index. In addition, Nle incorporation preserved the amphiphilic nature of [dLeu^9^, dLys^10^]TL because the value of the hydrophobic moment of [Nle^1^, dLeu^9^, dLys^10^]TL was similar to its lead peptide.

Moreover, the increase of hydrophobicity caused an enhancement of the antimicrobial activity for the peptide [Nle^1^, dLeu^9^, dLys^10^]TL both towards both Gram-positive and Gram-negative strains. In particular, the peptide displayed improved MIC values of 0.78 μM against *B. megaterium* Bm11 and 3.12 μM on *S. aureus* ATCC 25923 with respect to the higher values of the lead peptide. Interestingly, [Nle^1^, dLeu^9^, dLys^10^]TL resulted to be more active even on some Gram-negative strains with MIC of 3.12 against *A. baumannii* ATCC 19606 and MIC of 6.25 μM against *E. coli* ATCC 25922. However, this hydrophobic increase provoked variations of the cytotoxic profile of lead peptide [dLeu^9^, dLys^10^]TL both on human keratinocytes and murine macrophages at high tested concentrations. Indeed, while [dLeu^9^, dLys^10^]TL was not cytotoxic even at the highest concentration of 25 μM both on cell lines, the addition of Nle provoked a strong cytotoxic effect at 25 μM on keratinocytes and even at 10 μM on murine macrophages. As evidenced both from the physicochemical results and the biological data, the hydrophobicity influences the biological profile of [Nle^1^, dLeu^9^, dLys^10^]TL but also its behavior in the presence of the bacterial membrane. Herein, we studied the membrane interaction of [Nle^1^, dLeu^9^, dLys^10^]TL through biophysical studies using bio-membrane models as LUVs composed of DOPG/CL and DOPG/DOPE/CL that mimic, respectively, Gram-positive and Gram-negative membranes. The unbranched side chain of Nle promoted a monomeric form in solution calculating a CAC value of 38.9 μM, while the ThT assay provided a high propensity of [Nle^1^, dLeu^9^, dLys^10^]TL to oligomerize in DOPG/CL already at a low concentration of 5 μM, in contrast to the peptide [dLeu^9^, dLys^10^]TL that oligomerized progressively. This different behavior of [Nle^1^, dLeu^9^, dLys^10^]TL may be due to a better interaction with the membranes of Gram-positive bacteria that may also explain its better activity than its lead peptide. Instead, a better interaction was not evidenced with Gram-negative membrane, because we recorded a tendency of peptide [Nle^1^, dLeu^9^, dLys^10^]TL to oligomerize progressively as its lead peptide [dLeu^9^, dLys^10^]TL in presence of DOPG/DOPE/CL.

Since the binding of AMPs to the bacterial membrane determines a transfer of a random conformation into a range of secondary structures, including α-helices and β-structures, we also evaluated the conformational change of [Nle^1^, dLeu^9^, dLys^10^]TL in presence of DOPG/CL and DOPG/DOPE/CL by CD spectroscopy. We observed that the peptide forms β-aggregates when binding to and inserted into biological membranes, in contrast to lead peptide [dLeu^9^, dLys^10^]TL that forms α-helical aggregates. The change of conformation from aqueous solution to membrane mimetic environments is typical of AMPs, but it is more interesting to note the different structures attained in membranes just the incorporation of one amino acid hydrophobic amino acid residue.

Moreover, the repartition of both peptides in the interfacial region of membranes was also evidenced by the positive Wimley–White hydrophobicity values, indicating that the hydrophobic residues probably interact with phospholipids, while the hydrophilic residues are directed towards the lipid–water interface.

At this point, we explored the hypothetical mechanism of action of the peptide [Nle^1^, dLeu^9^, dLys^10^]TL in presence of LUVs through different biophysical techniques. Directly killing microbes and modulation of the immune system are the two main strategies that have been hypothesized for AMPs’ antimicrobial activity [57,58,59]. In our biophysical characterizations, we observed significant variations in membrane fluidity by calculating GP values after the treatment of LUVs with both peptides. In particular, we observed a strong variation of membrane fluidity of DOPG/CL after the treatment with 5 μM of [Nle^1^, dLeu^9^, dLys^10^]TL, detecting by a positive GP value of 0.03. This data highlighted the improved activity of [Nle^1^, dLeu^9^, dLys^10^]TL towards Gram-positive strains. Moreover, we also investigated the ability of peptide [Nle^1^, dLeu^9^, dLys^10^]TL to induce pore formation of the bacterial membrane. In our previous work, we observed a low capacity of peptide [dLeu^9^, dLys^10^]TL to induce leakage in presence of LUVs mimicking Gram-positive and Gram-negative membranes [27]. It is interesting to observe that the introduction of the unbranched residue Nle improved liposome leakage mimicking Gram-positive membrane, suggesting the pore formation and thus bacterial death. Meanwhile, we still observed a low leakage from liposome mimicking Gram-negative bacteria, assuming a carpeting effect or other molecular mechanisms for both peptides.

Finally, based on our previous study that demonstrated that the temporin L-analogue [dLeu^9^, dLys^10^]TL was able to induce a significant reduction of leukocyte infiltration and levels of inflammatory mediators in a model of zymosan-induce peritonitis [43]; herein, we also evaluated if the incorporation of Nle and increase of peptide hydrophobicity could also influence the anti-inflammatory activity of the peptide [dLeu^9^, dLys^10^]TL. We performed a preliminary in vitro study on J774A.1 cells where we observed a significant modulation of IL-6 production after the treatment with the peptide [Nle^1^, dLeu^9^, dLys^10^]TL at the concentration of 3 μM. This result shows that the Nle did not induce a loss of anti-inflammatory activity that could pave the way for future studies and in vivo investigations.

## 4. Materials and Methods

### 4.1. Peptide Synthesis

[dLeu^9^, dLys^10^]TL and [Nle^1^, dLeu^9^, dLys^10^]TL were synthesized by using the ultrasound-assisted solid-phase peptide synthesis (US-SPPS) in combination with the Fmoc/*t*Bu chemistry [27]. Rink amide resin (0.72 mmol/g as loading substitution) was chosen as solid support to have the *C*-terminal amide. Peptides were synthesized using repeated cycles of Fmoc deprotection using the solution of 20% piperidine in DMF (0.5 + 1 min) and coupling reactions. The latter reactions were accomplished by using Fmoc-amino acid (3 equiv) with HBTU (3 equiv), HOBt (3 equiv), DIPEA (6 equiv), dissolved in DMF, for 5 min under ultrasonic irradiations. After the sequence assembly, the N-terminal Fmoc protecting group was removed and peptides were released from the resin using a cocktail of trifluoroacetic acid/triisopropylsilane/H_2_O (95:2.5:2.5, *v:v:v*) for 3 h at room temperature. Then, crude peptides were purified by reversed-phase HPLC (RP-HPLC) using a preparative C-18 column and a gradient of acetonitrile and water containing 0.1% TFA (from 0 to 90% over 30 min).

### 4.2. Antimicrobial Susceptibility Testing

All Gram-negative and Gram-positive strains were obtained from the American Type Culture Collection (Rockville, MD, USA). MICs were determined as described previously [27]. 50 μL of bacterial culture properly diluted in Mueller–Hinton (MH) broth were added to each well of a 96-well plate containing 50 μL of MH with serially two-fold diluted peptide from 100 μM, to reach a final cell density of 1 × 10^6^ CFU/mL. Each MIC value indicates the lowest concentration of compound that causes a total inhibition of bacterial growth after 16–18 h of incubation at 37 °C.

### 4.3. Proteolytic Stability of [Nle^1^, dLeu^9^, dLys^10^]TL

The peptide [Nle^1^, dLeu^9^, dLys^10^]TL was incubated with human serum from human male AB plasma acquired by Sigma-Aldrich-Merk (Milan, Italy). The peptide was mixed with 90% of human serum at a final concentration of 200 μM and incubated at 37 °C [27]. Aliquots of the mixture were taken at time intervals of 15, 45, 75, 90 and 120 min. The serum protein was removed by the precipitation with acetonitrile (MeCN) and centrifuged at 13,000 rpm for 15 min. Then, the supernatant was taken and analyzed by ultra-high performance liquid chromatography (UHPLC) using a Phenomenex Kinetex column (4.6 mm × 150 mm, 5 μm, C18) and a linear gradient from 10 to 90% of MeCN (0.1% TFA) in water (0.1% TFA) over 20 min. The percentage of the intact peptide was calculated by integrating the peak area of the UHPLC chromatogram.

### 4.4. Hemolytic Assay

Hemolysis was evaluated on sheep red blood cells (OXOID, SR0051D, Milan, Italy). Aliquots of cells in 0.9% (*w/v*) NaCl were incubated with the peptide [Nle^1^, DLeu^9^, DLys^10^]TL at different concentrations (ranging from 3.12 to 100 μM) for 40 min at 37 °C with gentle shaking. Controls were erythrocytes treated with vehicle while total lysis controls were obtained by suspending cell in distilled water [27]. Then, the samples were centrifuged for 5 min at 900× *g*, and the presence of hemoglobin in the supernatant was measured at an absorbance of 415 nm using a microplate reader (Infinite M200; Tecan, Salzburg, Austria) and compared to the complete lysis of erythrocytes (100%). The data reported are the mean ± the standard error of the mean (S.E.M.) of three independent experiments.

### 4.5. Cytotoxicity on Human Keratinocytes

The human immortalized keratinocytes HaCaT cells were purchased from AddexBio (San Diego, CA, USA) and cultured in Dulbecco’s modified Eagle’s medium supplemented with 4 mM glutamine (DMEMg), 10% heat-inactivated fetal bovine serum (FBS), and 0.1 mg/mL of penicillin and streptomycin, at 37 °C and 5% CO_2_, in 25 cm^2^ or 75 cm^2^ flasks. The in vitro MTT [3-(4,5-dimethylthiazol-2-yl)-2,5-diphenyltetrazolium bromide] assay, based on the reduction of the yellow tetrazolium salt MTT to the purple formazan by viable cells with active metabolism, was used to evaluate the peptide cytotoxicity on HaCaT cells, as previously described in [27]. Specifically, 4 × 10^4^ HaCaT cells, suspended in DMEMg supplemented with 2% FBS were plated in a 96-well microtiter plate. After overnight incubation, at 37 °C in a 5% CO_2_ atmosphere, HaCaT cells were treated for 2 h or 24 h with fresh serum free medium containing the peptide [Nle^1^, DLeu^9^, DLys^10^]TL in a concentration range from 3.12 to 100 μM. Then, the medium was discarded and 0.5 mg/mL of MTT in Hank’s buffer was added to each well, incubating the plate for further 4 h. Formazan crystals were dissolved using acidified isopropanol, and the absorbance of each well was measured at 570 nm using a microplate reader (Infinite M200; Tecan, Salzburg, Austria). The number of viable cells was proportional to the intensity of purple color and was expressed as percentage compared to that of untreated control cells (100%). The data are the mean ± S.E.M. of three independent experiments.

### 4.6. Peptide Aggregation in Solution and in LUVs

Peptide aggregation was monitored in aqueous solution and in large unilamellar vesicles (LUVs) made of 1,2-dioleoyl-sn glycero-3-phospho-(10-rac-glycerol) sodium salt (DOPG) and cardiolipin (CL) sodium salt (Heart, Bovine) mimicking Gram-positive membrane, and DOPG, CL and 1,2-dioleoyl-sn-glycero-3-phosphoethanolamine (DOPE), mimicking Gram-negative membrane.

Nile Red (NR) was used as fluorescent probe to monitor the aggregation in solution [51]. NR is not very soluble in water and has a large preference to move into hydrophobic environment as aggregates, producing a blue shift and hyperchromic effect. The peptide stock (400 μM) was prepared in 1,1,1,3,3,3-hexafluoro-2-propanol (HFIP) as solvent and different aliquots were taken to prepare each peptide solution at different concentrations (1, 5, 10, 15, 20, 30, 50, 100, and 200 μM) [60]. Then, HFIP was removed under nitrogen stream, 1 mL of water was added, and the solution was sonicated for 15 min and freeze-dried. Finally, the peptide powders were dissolved with a solution of 500 nM of NR in water and equilibrated for 1 h at room temperature. Emission spectra for each solution were measured between 570 and 700 nm by a Cary Eclipse Varian spectrometer (San Diego, CA, USA), setting a slit width of 5 nm and an excitation wavelength of 550 nm and a 10 nm slit width. The measurements were performed in triplicate. The data were analyzed by plotting the maximum emission fluorescence corresponding wavelength (y) as a function of peptide concentration (x) and fitting with the sigmoidal Boltzmann equation as described previously [27,60].

Regarding peptide aggregation in LUVs, was monitored using the Thioflavin T as fluorescent probe [61]. LUVs were prepared using the extrusion method. Firstly, the lipid films (final concentration of 0.1 mM), made of DOPG, CL (58/42 ratio in moles) and DOPG/DOPE/CL (63/23/12 ratio in moles), were prepared by dissolving an appropriate amount of lipids in chloroform. The solvent was removed under nitrogen stream, water was added, and the film lipids were freeze-dried. Then, the lipid films were hydrated with 100 mM NaCl, 10 mM Tris-HCl, 25 μM Tht buffer, pH 7.4, and vortexed for 1 h. Then, the lipid suspension was freeze-thawed 6 times and extruded 10 times through polycarbonate membranes with 0.1 mm diameter pores to obtain LUVs [27]. The aggregation of [Nle^1^, DLeu^9^, DLys^10^]TL was monitored titrating LUVs with different peptide concentrations of 5, 10, 15, 20, 30, 50 μM. Spectra were recorded before and after the addition of peptide using a Varian Cary Eclipse fluorescence spectrometer, exciting the sample at 450 nm (slit width, 10 nm) and recording fluorescence emission at 482 nm (slit width, 5 nm). Aggregation was quantified according to this equation: %A= (F_f_ × F_0_)/(F_max_ × F_0_) × 100, where F_f_ indicates the value of fluorescence after peptide addition, F_0_ the fluorescence in the absence of peptide and F_max_ is the fluorescence maximum obtained immediately after peptide addition.

### 4.7. Circular Dichroism in SUVs

All CD spectra were recorded from 260 to 190 nm (a bandwidth of 3 nm, a time constant of 16 s, and a scan rate of 10 nm/min) on a Jasco J-175 spectropolarimeter using a 1 cm quartz cell at 25 °C. Small unilamellar vesicles (SUVs) made of DOPG/CL (58/42 ratio in moles) and DOPG/DOPE/CL (63/23/12 ratio in moles) were used in the conformational study. SUVs were prepared as reported previously [27]. Firstly, lipids dissolved in chloroform and an equal volume of peptide solution dissolved in 2,2,2-trifluoroethanol (TFE) containing the peptide [Nle^1^, dLeu^9^, dLys^10^]TL at a concentration of 3 μM were mixed, vortexed, and lyophilized overnight. Then, the dry samples were hydrated with phosphate buffer 5 mM, pH 7.4, for 1 h and sonicated for 30 min to obtain SUVs [62]. Each spectrum was corrected for the blank and converted into mean molar ellipticity.

### 4.8. Membrane Fluidity

Membrane fluidity was evaluated using LUVs mimicking Gram-positive and Gram-negative membrane and Laurdan as fluorescent probe. Laurdan at a concentration of 0.001 mM was encapsulated in lipid films at a final concentration of 0.1 mM. After the preparation of lipid films, they were hydrated with PBS 1× buffer, pH 7.4, and treated using the extrusion method to obtain LUVs. The variation of membrane fluidity was evaluated treating LUVs with two different peptide concentrations of 5 μM and 30 μM. Each measurement was performed recording Laurdan emission spectra from 400 to 550 nm with λ_ex_ 365 nm both in the absence and presence of peptide. Laurdan emission can shift from 440 nm, in the ordered phase to 490 nm in the disordered phase [52]. The change in membrane fluidity was measured by calculating Generalized Polarization (GP) parameter. GP is calculated as GP = (I_440_ − I_490_)/(I_440_ + I_490_), where I_440_ and I_490_ indicate the fluorescence intensities at the maximum emission wavelength in the ordered and disordered, respectively.

### 4.9. ANTS/DPX Leakage Assay

The ability of the peptide [Nle^1^, dLeu^9^, dLys^10^]TL to induce LUVs leakage was evaluated using ANTS/DPX assay [63]. After the preparation of lipid films (0.1 mM) as described above, fluorophores ANTS (12.5 mM) and DPX (45 mM) were dissolved in water and added to lipid films. The mixture was frozen and lyophilized overnight. Then, the lipid films with ANTS and DPX were hydrated with PBS 1× buffer, vortexed for 1 h and then treated to obtain LUVs. The nonencapsulated ANTS and DPX were removed by gel filtration using a Sephadex G-50 column (1.5 cm × 10 cm) at room temperature. The leakage experiment was performed titrating LUVs with different peptide concentrations of 5, 10, 15, 20, 30, 50 μM, and each sample was excited at 385 nm (slit width, 5 nm) and the fluorescence emission was recorded at 512 nm (slit width, 5 nm). The liposomes leakage promoted by peptide was correlated with an increase of ANTS fluorescence at 512 nm. In this experiment, the control was represented by the complete release of ANTS after the treatment of LUVs with 0.1% Triton. The liposomes leakage was calculated as % Leakage = (F_i_ − F_0_)/(F_t_ − F_0_), where F_0_ is the fluorescence of intact LUVs before the addition of peptide, and F_i_ and F_t_ is the intensities of the fluorescence achieved after peptide and Triton-X treatment, respectively.

### 4.10. Peptide Cytotoxicity on Murine Macrophage J774A.1 Cells

To evaluate the effect of [Nle^1^, dLeu^9^, dLys^10^]TL on J774A.1 murine macrophages, cell viability was examined as previously described [43]. Briefly, using a colorimetric assay based on the MTT labelling reagent, J774A.1 cells (25 × 10^3^ per well) were seeded in 96-well plates and, after overnight incubation, were treated with [Nle^1^, dLeu^9^, dLys^10^]TL at concentrations of 1, 3, 10, and 30 μM. The spectrophotometric absorbance was then measured using a microtiter enzyme-linked immunosorbent assay reader (Multiskan™ GO Microplate Spectrophotometer; Thermo Scientific™, Milan, Italy) at 540 nm. The percentage of cell viability was assessed by the following formula: (OD of treated cells/OD of control) × 100.

### 4.11. Anti-Inflammatory Activity on Murine Macrophages

Mouse macrophage cell line J774A.1 was cultured as previously described. Cells were seeded in Petri culture dishes (100 mm × 20 mm) at a density of 5 × 10^5^ cells per dish and allowed to grow for 24 h. The medium was then replaced, and cells were treated with LPS (10 μg∙mL^−1^) in presence or absence of [Nle^1^, dLeu^9^, dLys^10^]TL at the concentration of 3 µM (non-cytotoxic concentration from MTT assay). Following incubation of 24 h, cells were collected with a cell scraper, and after centrifugation at 14,000× *g* for 10 min at 4 °C, the supernatant was collected and assayed for IL-6 enzyme-linked immunosorbent assay (ELISA) analysis [43].

### 4.12. Data and Statistical Analysis

All data are presented as means ± S.D. or S.E.M. and were analysed using one-way ANOVA followed by Bonferroni’s for multiple comparisons. GraphPad Prism 8.0 software (San Diego, CA, USA) was used for analysis. Differences among groups were considered significant when *p* ≤ 0.05 was achieved according to international recommendations on experimental design and analysis in preclinical pharmacology [64,65].

## 5. Conclusions

In conclusion, here, we increased the hydrophobicity of the peptide [dLeu^9^, dLys^10^]TL introducing a norleucine residue in its flexible *N*-terminal region, because its unbranched side chain may favor the insertion into the lipid bilayer of bacterial membranes.

The biological data showed that the enhancement of hydrophobicity was correlated with an increase in antimicrobial activity of the peptide [Nle^1^, dLeu^9^, dLys^10^]TL both against Gram-positive and Gram-negative strains. The correct hydrophobic/hydrophilic balance of the lead peptide was preserved because [Nle^1^, dLeu^9^, dLys^10^]TL did not display cytotoxicity at its antimicrobial concentration. Biophysical studies on the mode of action of [Nle^1^, dLeu^9^, dLys^10^]TL highlighted the formation of β-aggregates in bacterial membranes with a pore formation in presence of LUVs mimicking Gram-positive membrane. Instead, in presence of LUVs mimicking Gram-negative membrane, we hypothesized a carpeting effect or other molecular mechanisms for the peptide [Nle^1^, dLeu^9^, dLys^10^]TL. Moreover, it is noteworthy that the increase of hydrophobicity promoted a better anti-inflammatory activity for [Nle^1^, dLeu^9^, dLys^10^]TL in a preliminary in vitro model.

## 6. Patents

The described peptide is the object of a patent application (Number application: 102021000029738), in date: 24 November 2021.

## Figures and Tables

**Figure 1 antibiotics-11-01285-f001:**
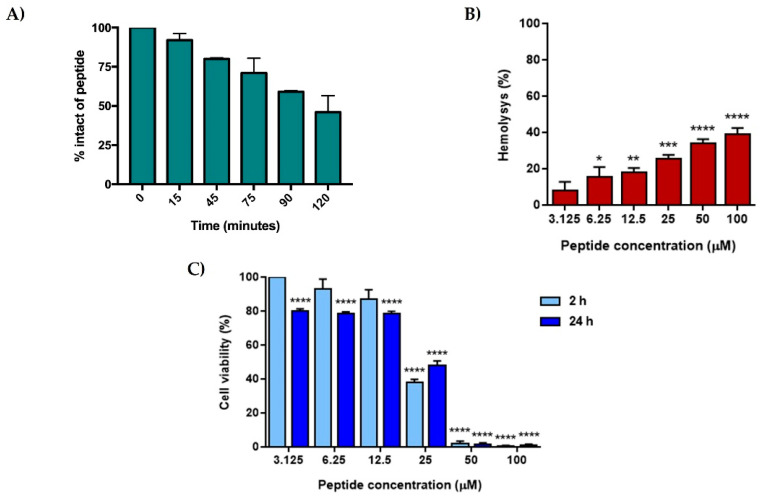
Percentage of intact peptide after the incubation of fresh human serum (panel **A**), and the hemolysis and cytotoxicity of peptide [Nle^1^, dLeu^9^, dLys^10^]TL on red blood cell (panel **B**) and human keratinocytes (panel **C**). For panel **B** and **C**, values are presented as means ± S.E.M. of 3 independent experiments. * *p* < 0.05, ** *p* < 0.01, *** *p* < 0.001, **** *p* < 0.0001 vs. Ctrl group (0% and 100% for **B** and **C**, respectively). Statistical analysis was performed by one-way ANOVA followed by Bonferroni’s multiple comparisons test.

**Figure 2 antibiotics-11-01285-f002:**
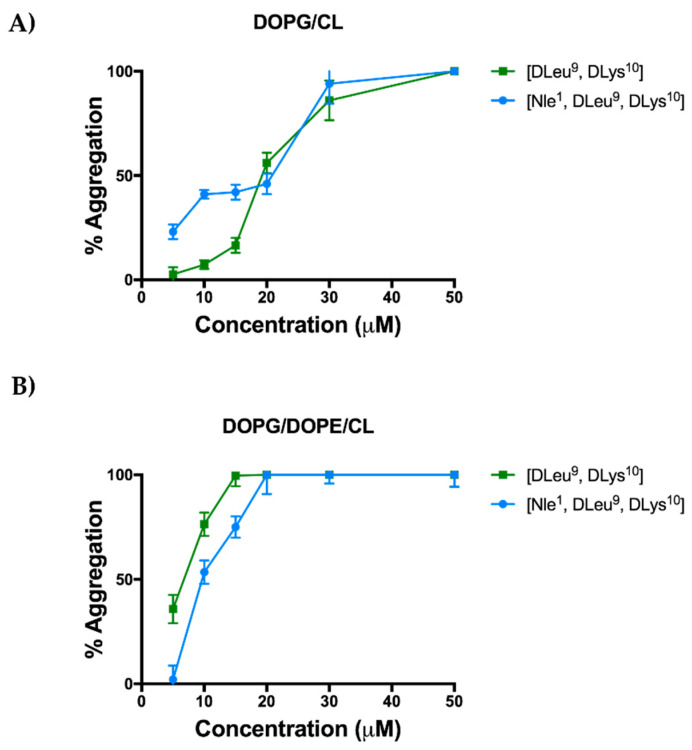
Peptide aggregation monitored by ThT in presence of LUVs mimicking Gram-positive (DOPG/CL, panel **A**) and Gram-negative (DOPG/DOPE/CL, panel **B**) cell membranes.

**Figure 3 antibiotics-11-01285-f003:**
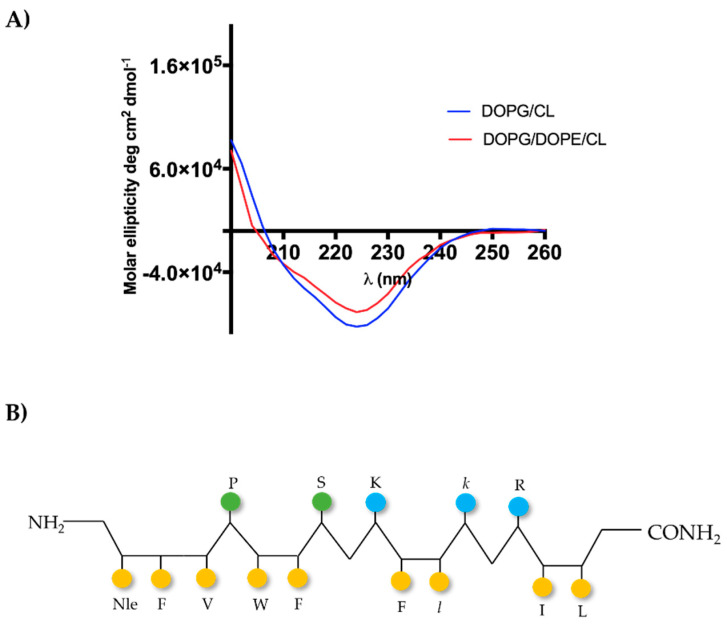
The panel A shows CD spectra of [Nle^1^, dLeu^9^, dLys^10^]TL in LUVs made of DOPG/CL and DOPG/DOPE/CL (panel **A**). The (panel **B**) shows the hypothetical representation of β-sheet conformation of [Nle^1^, dLeu^9^, dLys^10^]TL, where yellow circles indicate hydrophobic residues, blue ones positively charged amino acids, and green the other amino acids.

**Figure 4 antibiotics-11-01285-f004:**
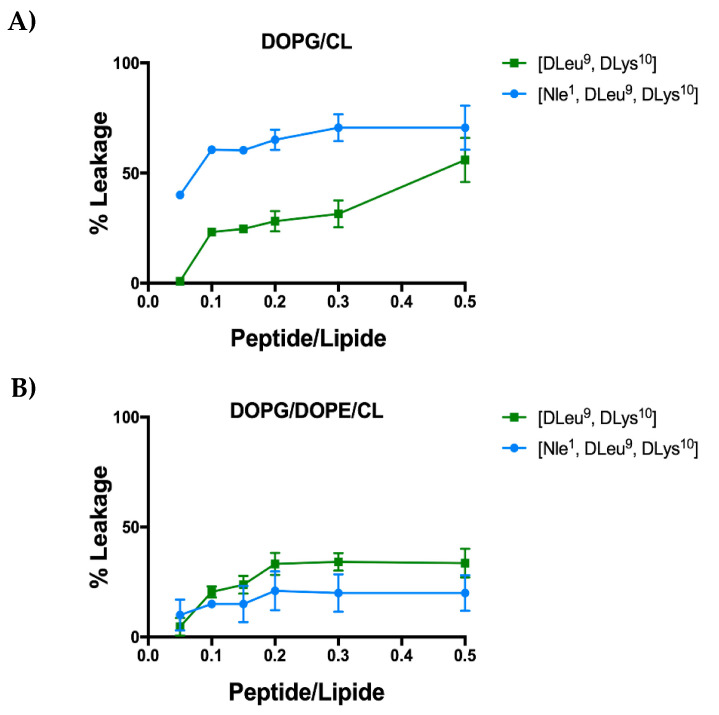
Leakage of DOPG/CL (panel **A**) and DOPG/DOPE/CL (panel **B**) after the treatment with the peptide [Nle^1^, dLeu^9^, vLys^10^]TL. The leakage data of [dLeu^9^, dLys^10^]TL were already reported in Bellavita et al. [27].

**Figure 5 antibiotics-11-01285-f005:**
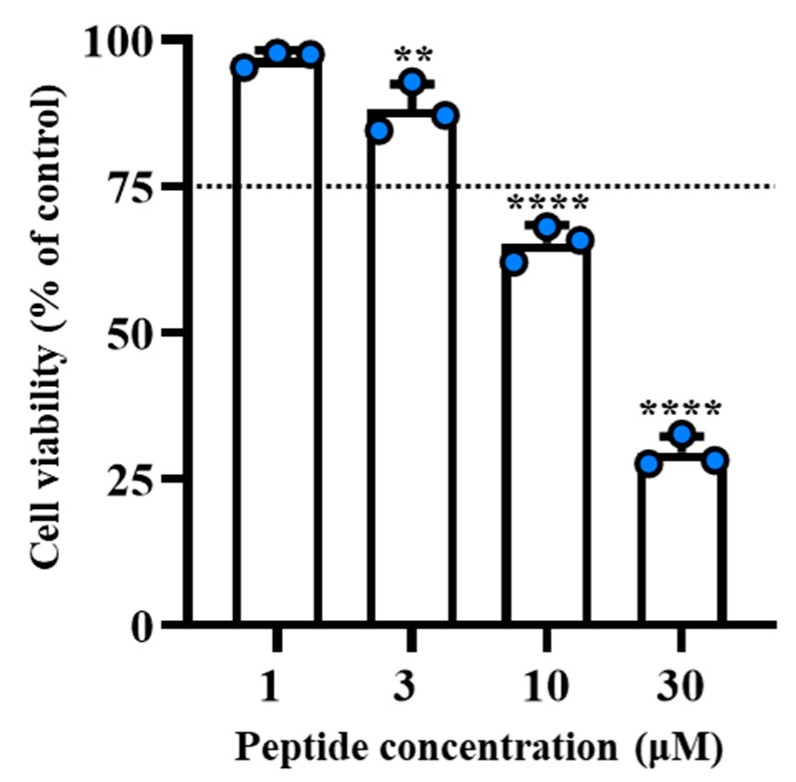
In vitro cytotoxic examination for [Nle^1^, dLeu^9^, dLys^10^] at concentrations of 1–30 μM was performed on J774A.1 murine macrophage cells. Dotted lines indicate the 75% of cell viability. Values are presented as means ± S.D. of 3 independent experiments. ** *p* ≤ 0.01, **** *p* < 0.0001 vs. Ctrl group (100%). Statistical analysis was performed by one-way ANOVA followed by Bonferroni’s for multiple comparisons.

**Figure 6 antibiotics-11-01285-f006:**
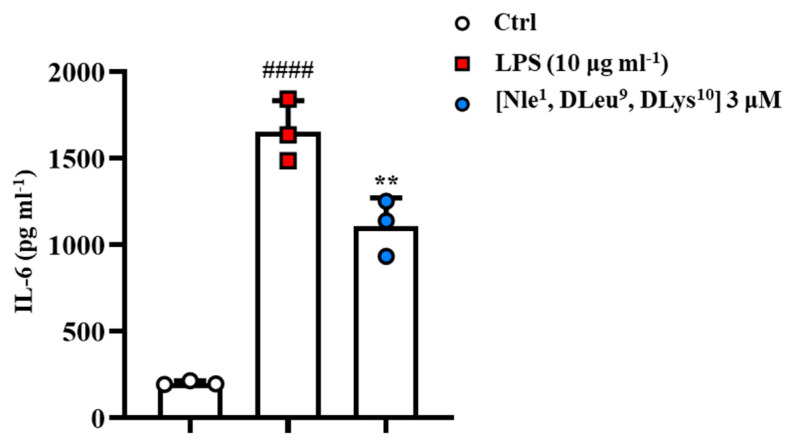
J774A.1 murine macrophage cells were stimulated with LPS (10 µg mL^−1^) and treated with [Nle^1^, dLeu^9^, dLys^10^] at the concentration of 3 μM for 24 h. Thereafter, cell supernatants were assayed for IL-6 enzyme linked immunosorbent assay as pg mL^−1^ and presented as means ± S.D. of three independent experiments. ^####^ *p* ≤ 0.0001 vs. Ctrl group; ** *p* ≤ 0.01 vs. LPS group. Statistical analysis was performed by one-way ANOVA followed by Bonferroni’s for multiple comparisons.

**Table 1 antibiotics-11-01285-t001:** Physicochemical properties of [Nle^1^, dLeu^9^, dLys^10^]TL in comparison with its peptide [dLeu^9^, dLys^10^]TL.

Peptide	GRAVY	Aliphatic Index	Hydrophobicity *	Hydrophobic Moment (μH)
[dLeu^9^, dLys^10^]TL	0.700	112.31	+4.15 Kcal∙mol^−1^	0.670
[Nle^1^, dLeu^9^, Lys^10^]TL	0.921	132.14	+2.90 Kcal∙mol^−1^	0.601

* Hydrophobicity (Wimley–White scale).

**Table 2 antibiotics-11-01285-t002:** MICs values of peptide [Nle^1^, dLeu^9^, dLys^10^]TL in comparison with [dLeu^9^, dLys^10^]TL on selected Gram-positive and Gram-negative strains.

Strains	MIC Values (μM)	
	[Nle^1^, dLeu^9^, dLys^10^]TL	[dLeu^9^, dLys^10^]TL *
*E. coli* ATCC 25922	6.25	12.5
*P. aeruginosa* ATCC 27853	12.5	12.5
*A. baumannii* ATCC 19606	3.12	6.25
*K. pneumoniae* ATCC BAA-1705	12.5	12.5
*S. aureus* ATCC 25923	3.12	6.25
*S. epidermidis* ATCC 12228	3.12	3.12
*B. megaterium* Bm11	0.78	3.12

* The MIC values for the lead peptide were already reported in previous works [27,41].

**Table 3 antibiotics-11-01285-t003:** GP values calculated as GP = (I_440_ − I_490_)/(I_440_ + I_490_) before and after the treatment with the peptide [Nle^1^, dLeu^9^, dLys^10^]TL. The GP values of lead peptide [dLeu^9^, dLys^10^]TL was calculated previously [27].

GP Value			
LUVs: DOPG/CL			
cmpd	Unloaded LUVs	LUVs + 5 μM cmpd	LUVs + 30 μM cmpd
[dLeu^9^, dLys^10^]TL	−0.18 ± 0.01	−0.06 ± 0.01	0.26 ± 0.02
[Nle^1^, dLeu^9^, dLys^10^]TL	−0.18 ± 0.01	0.03 ± 0.01	0.29 ± 0.03
LUVs: DOPG/DOPE/CL			
cmpd	Unloaded LUVs	LUVs + 5 μM cmpd	LUVs + 30 μM cmpd
[dLeu^9^, dLys^10^]TL	−0.07 ± 0.01	0.09 ± 0.02	0.11 ± 0.02
[Nle^1^, dLeu^9^, dLys^10^]TL	−0.07 ± 0.01	0.05 ± 0.01	0.13 ± 0.03

## Data Availability

Not applicable.

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
