# Peer review of "Synthetic Amphipathic β-Sheet Temporin-Derived Peptide with Dual Antibacterial and Anti-Inflammatory Activities"

_antibiotics, 2022, doi:10.3390/antibiotics11101285_

Round 1

Reviewer 1 Report

1. The Results section reflects the authors' results and should avoid describing how they were achieved, which is appropriate for the Materials and Methods  section (2.1. Peptide Design and Synthesis). Other parts are too descriptive and are suitable for Introduction rather than for Results.

2. I did not found any statistical analyses and hence statistically significant differences about subsections from 2.2 to 2.7 that make the results and conclusions debatable.

3. Discussion part should be in context with your achievements, explain only your important results.

Author Response

1. The Results section reflects the authors' results and should avoid describing how they were achieved, which is appropriate for the Materials and Methods section (2.1. Peptide Design and Synthesis). Other parts are too descriptive and are suitable for Introduction rather than for Results.

Reply: We thank the reviewer for his/her comment. We modified the results section accordingly.

2. I did not found any statistical analyses and hence statistically significant differences about subsections from 2.2 to 2.7 that make the results and conclusions debatable.

Reply: We thank the reviewer for his/her comment. We modified figure 1 and table 3 in the main text, accordingly.

3. Discussion part should be in context with your achievements, explain only your important results.

Reply: We thank the reviewer for his/her comment. We modified the discussion part accordingly.

Reviewer 2 Report

The present manuscript, ''Synthetic amphipathic-sheet temporin-derived peptide with dual antibacterial and anti-inflammatory activities'', is a high-quality paper that provides valuable insights for the research and development of innovative and reliable antimicrobial solutions, addressing a vitally important need of the pharmaceutical industry.

However, I have a couple of suggestions for the authors, as mentioned below:

  • lines 164-177: Could you please mention the reference antibacterial drug used in the microbiological assay? Additionally, could you please add an analogy between the obtained results for the tested peptide and those obtained for the reference antibacterial drug?

Author Response

Lines 164-177: Could you please mention the reference antibacterial drug used in the microbiological assay? Additionally, could you please add an analogy between the obtained results for the tested peptide and those obtained for the reference antibacterial drug?

Reply: We thank the reviewer for his/her comment. The aim of the work was to test the antimicrobial activity of an AMP, which belongs to an already well-characterized peptide family (i.e. Temporins). Hence, we did not use a conventional reference antibiotic in the assay, but instead, we used a temporin-derived peptide with a known activity (i.e., [Pro3,dLeu9]TL  (Bellavita et al., J. Med. Chem. 2021, 64, 11675-11694, doi: 10.1021/acs.jmedchem.1c01033) as an internal control of the experiments.

Reviewer 3 Report

In this Manuscript, Bellavita and co-authors studied the antimicrobial activity and anti-inflammatory activity of modified Temporin-L by introducing Norleucine at N-terminus. From their physiochemical property studies (Table 1.), they found increased hydrophobic nature of peptides that influence the antimicrobial activity in comparison with lead temporin (Table 2). The author was able to show better results in comparison with lead temporin peptide by simple modification using Norleucine. However, the author has a better chance to prepare several of its analogs and study their activities. Even this single modification has impactful results from there several physiochemical studies and antimicrobial testing on several gram-positive and gram-negative bacteria.  This indicates that in the near future these modified AMPs could be a potential candidate for further preclinical development of the novel anti-microbial and anti-inflammatory activity.  The author was also able to study their improved anti-inflammatory activity after introducing norleucine amino acid to temporin.

Besides this, the manuscript is well-written, the content is explained scientifically, and the data support the hypotheses. Based on the potential and strength of the present research, I would recommend it for publication in the Antibiotics without changes.

Author Response

We thank the reviewer for his/her comment.

Round 2

Reviewer 1 Report

The authors have made an effortя to clean up much of the remarks, although there are still sentences in the Results section that I think sound descriptive.